# Characterization of a Novel 2018 Influenza Virus Outbreak on the Yucatan Peninsula, Mexico, in the Summer

**DOI:** 10.3390/microorganisms13051086

**Published:** 2025-05-07

**Authors:** Lumumba Arriaga-Nieto, David Alejandro Cabrera-Gaytán, Alfonso Vallejos-Parás, Porfirio Felipe Hernández-Bautista, Clara Esperanza Santacruz-Tinoco, Julio Elías Alvarado-Yaah, Yu-Mei Anguiano-Hernández, Bernardo Martínez-Miguel, María Erandhí Prieto-Torres, Concepción Grajales-Muñiz, Nancy Sandoval-Gutiérrez, Horacia Celina Velarde-Scull

**Affiliations:** 1Coordinación de Vigilancia Epidemiológica, Instituto Mexicano del Seguro Social, Mexico City 03100, Mexico; lumumba.arriaga@imss.gob.mx (L.A.-N.); alfonso.vallejos@imss.gob.mx (A.V.-P.); 2Coordinación de Calidad de Insumos y Laboratorios Especializados, Instituto Mexicano del Seguro Social, Mexico City 07760, Mexico; porfirio.hernandez@imss.gob.mx (P.F.H.-B.); clara.santacruz@imss.gob.mx (C.E.S.-T.); julio.alvaradoy@imss.gob.mx (J.E.A.-Y.); yu.anguiano@imss.gob.mx (Y.-M.A.-H.); bernardo.martinezm@imss.gob.mx (B.M.-M.); nancy.sandovalg@imss.gob.mx (N.S.-G.); 3Órgano de Operación Administrativa Desconcentrada en Quintana Roo, Instituto Mexicano del Seguro Social, Chetumal 77530, Mexico; erandhi.prieto@imss.gob.mx; 4Organismo Público Descentralizado Servicios de Salud del Instituto Mexicano del Seguro Social para el Bienestar (IMSS-BIENESTAR), Mexico City 01030, Mexico; concepcion.grajales@imssbienestar.gob.mx; 5Unidad Médica de Alta Especialidad, Hospital de Pediatría del Centro Médico Nacional de Occidente, Instituto Mexicano del Seguro Social, Guadalajara 44329, Mexico; horacia.velarde@imss.gob.mx

**Keywords:** novel influenza A, influenza A (H1N1)pdm09, influenza-like illness, severe acute respiratory infection

## Abstract

During the 2017–2018 influenza season, there was high influenza activity, with a predominance of influenza A(H1N1)pdm09 circulation in the country. The influenza circulation pattern in the area of the Yucatan Peninsula was different from that of the rest of the country. However, in the summer of 2018, there was a sudden increase in the number of identified cases. A retrospective analysis was performed using data generated by four molecular diagnostic laboratories of the Mexican Social Security Institute. Demographics, influenza positivity, seasonality and case fatality rates were recorded. We used odds ratios to compare outpatients who were confirmed by laboratory tests to be positive with those who were confirmed to be negative. The Kaplan–Meier method and Cox multivariate analysis were used to calculate cumulative risk. There were 4460 cases of ILI/SARI between Yucatan and Quintana Roo, which represented 53.1% of the total number of cases reported. Compared with that in 2009, the epidemic wave in 2018 was shorter and more expansive, with a greater number of reported cases, as well as a greater number of people who required hospitalization. The dominant pattern of A(H1N1)pdm09 influenza activity on the Yucatan Peninsula in the summer of 2018 has not been observed since the influenza pandemic of 2009.

## 1. Introduction

In 2009, the world faced the first influenza pandemic of the 21st century, and it was caused by the influenza A(H1N1)pdm09 virus; reports of respiratory hospitalizations and deaths among young adults in Mexico alerted local health officials to the atypical rates of respiratory illness at a time when influenza was not expected to reach epidemic levels [1]. Subsequently, the influenza A(H1N1)pdm09 virus began to appear seasonally in the following years, cocirculating with other influenza viruses [2,3]. Therefore, epidemiological virological surveillance is important because it allows for the detection of circulating strains and transmission patterns of influenza and the identification of the groups within the population that are most vulnerable to influenza.

The influenza A(H1N1)pdm09 virus circulates in a biennial pattern in Mexico [3]. During the 2016–2017 seasonal period, the epidemiological surveillance system of the Institute was able to identify the circulation of the influenza A(H1N1)pdm09 virus in advance; given that, according to the historical behavior of the genotype, it was expected that influenza A(H1N1)pdm09 would have low circulation due to influenza A(H3), the 2016–2017 winter season was considered to have high virus circulation, where the highest concentration of cases was in the northern region of the country [2,4].

During the 2017–2018 influenza season, there was high influenza activity, with a predominance of influenza A(H1N1)pdm09 circulation in the country. The states of the Yucatan Peninsula had different influenza activity compared to the rest of the country, with the beginning of the epidemic wave occurring between August and September [4]. However, in the interseasonal period of 2018, there was a sudden increase in the number of identified cases. Additionally, this was the first time that an abrupt increase in influenza circulation has been documented, which was greater than what was observed during the 2009 influenza pandemic. We described a substantial recrudescent wave of influenza A(H1N1)pdm09 affecting the population of the Yucatan Peninsula with social security during the summer of 2018.

## 2. Materials and Methods

### 2.1. Data Collection and Surveillance

The Mexican Institute for Social Security (Instituto Mexicano del Seguro Social, IMSS) provides active surveillance for influenza-like illness in outpatients within all primary healthcare units and in inpatients and mortality surveillance in hospitals. Patients who visited any primary care clinic or hospital and met the case definition of influenza-like illness (ILI) and those who were admitted with such an illness were entered into the surveillance system online by hospital or clinic epidemiologists, and the database was updated every day. ILI was defined as fever, cough, and headache, with one or more of the following symptoms: sore throat, rhinorrhea, arthralgias, myalgia, prostration, thoracic pain, abdominal pain, nasal congestion or diarrhea, and irritability in infants (fever was not needed as a symptom for people older than 65 years). A person of any age was considered to have a severe acute respiratory infection (SARI) if they presented with difficulty breathing accompanied by a fever greater than or equal to 38 °C and a cough with one or more of the following symptoms: effects on general condition, chest pain, polypnea or acute respiratory distress syndrome, or death associated with SARI. In immunosuppressed patients and those older than 65 years, fever is not a cardinal symptom. Children <5 years of age with pneumonia or severe pneumonia that required hospitalization were also considered SARI patients; these operational definitions are used in a standardized way throughout the country [5].

The requested information for case notification included age, sex, occupation, symptoms, and the presence of a chronic disease from the Online Epidemiological Surveillance System (SINOLAVE), which is an epidemiological surveillance platform implemented by the IMSS during the 2009 influenza pandemic in Mexico to collect information about patients with acute respiratory infection seeking care at IMSS healthcare facilities. For each patient, SINOLAVE collects data related to demographic characteristics (age, sex), underlying conditions (self-reported chronic diseases), symptoms at the time of testing, dates of symptom onset (self-reported), influenza laboratory results and output. We constructed weekly epidemic curves describing temporal patterns of confirmed cases and positivity by week.

### 2.2. Diagnostic Tests

All the tests were performed according to the national instructions and laboratory-validated protocols at the Molecular Diagnostic Laboratory for Influenza, Yucatan. From the total number of samples tested, the following results were available for determining the performance of the diagnostic tests for the detection of influenza A viruses via quantitative reverse transcription quantitative polymerase chain reaction (RT-qPCR) [6]. According to the guidelines of the General Directorate of Epidemiology, the samples sent represent 10% of outpatient cases and 100% of hospitalized cases [5,6].

Laboratory confirmation was performed following the protocol described by the World Health Organization (WHO) [7] for the confirmatory diagnosis of influenza A(H1N1)pdm09, A/H3 and influenza B. Pharyngeal swab samples from the epidemiological surveillance system were processed by extracting nucleic acids using the automated MagNa Pure LC 2.0 equipment, ROCHE or the manual extraction method QIAamp VIRAL RNA, QIAGEN, following the manufacturer’s recommendations, and analyzed by RT-qPCR using the 7500 fast-applied biosystems thermal cycler (Thermo Fisher Scientific). The assay used for the identification of the influenza virus was the WHO protocol for polymerase chain reaction prior to reverse transcription of the viral genome (RT-qPCR) with the SuperScript III Reverse Transcriptase/Platinum TaqDNA polymerase reagent (Invitrogen), which reacts with the M gene (matrix protein, InfA), NA gene (influenza A(H1N1)pdm09, influenza A H3N2), HA gene (influenza A(H1N1)pdm09, influenza B Victoria and Yamagata lineages) and human ribonuclease P as an endogenous control.

### 2.3. Statistical Analysis

We analyzed information collected by the influenza surveillance system from weeks 21 to 37 of 2018 for patients with ILI and SARI who visited clinics that were part of the Mexican Institute for Social Security network. We estimated rates of clinical infection within patients with a family doctor as the denominator. To identify symptoms with an increased association with testing positive for influenza infection and demographic characteristics, we used odds ratios (ORs) and 95% CIs to compare outpatients who were laboratory-confirmed positive cases with those confirmed negative, with unconditional logistical regression adjusted for age and sex. The Wilson score was calculated with 95% confidence limits for proportions in Open Source Epidemiologic Statistics for Public Health. Bivariate analysis was also performed to estimate risks using the Epi Info v 7.2.4,0, R v 4.1.1.1. For deaths, the data were analyzed using survival analysis methods; survival was estimated with standardized cumulative risk curves. The Kaplan–Meier method and Cox multivariate analysis were used to calculate cumulative risk. Additionally, the case fatality rate was estimated.

## 3. Results

### 3.1. Overall Epidemiological Patterns

During 2018, 4460 cases were identified between Yucatan and Quintana Roo, of which 573 were confirmed by laboratory tests (12.8%), whereas during the 2009 influenza pandemic, 4702 cases were identified, with 1431 cases confirmed by laboratory tests (30.4%). The 4460 cases of ILI/SARI represented 53.1% of the total number of cases reported in the IMSS during week No. 20 to 37 of 2018.

The time series of weekly ILI/SARI cases and laboratory-confirmed influenza cases are shown in Figure 1. The duration of the epidemic waves differed between 2009 and 2018; in the first wave, it was 30 weeks (weeks 9 to 38), whereas in the second wave, it was 17 weeks (weeks 21 to 37). The incidence rates of ILI/SARI were 271.1 in 2009 and 250.4 per 100,000 people with a family doctor (Appendix A, Table A1 and Table A2).

### 3.2. Age Patterns

The majority of confirmed cases occurred in the 25–44 years age group in 2018 (42.7%), whereas in 2009, the age group was 15–24 years (Appendix A, Table A3).

During 2018, the majority of reported ILI/SARI cases were in Yucatan, with 3768, and Quintana Roo, with 692. A notable difference across age groups was that in Quintana Roo, cases in those 65 and older accounted for 11.6%, while in Yucatan, the percentage was 3.7%. Similarly, in children under four years of age, the percentage was higher in Quintana Roo (13.9% vs. 8.4%). However, when estimating the incidence rate per 100,000 people affiliated with family doctors, Yucatan represented the highest incidence overall and by age group (Appendix A, Table A1 and Table A2).

### 3.3. Geographical Extension

The largest proportion of cases reported to the epidemiological surveillance system was in Yucatan (44.9%) and Quintana Roo (8.3%). The majority of cases during the wave were reported in Mérida (76.2%) in Yucatan and Cancun (64.7%) in Quintana Roo, and a greater proportion of influenza A(H1N1)pdm09 deaths occurred in the same localities in 2018.

### 3.4. Clinical Patterns and Severity of Disease

The risk factor for influenza illness was mainly a history of contact with animals (OR = 5.08, 95% CI 3.53 to 7.26), whereas in 2009, it was asthma (OR = 1.86, 95% CI 1.32 to 2.63), although it remained an associated factor in both periods (Table 1). The symptoms associated with confirmed cases were cough, fever, headache and prostration, with sudden onset, both during the outbreak in 2018 and in 2009. When a multivariate model was constructed with the risk factors and symptoms, fever, cough and coryza remained statistically significant (Table 2).

The patients who presented with a chronic disease presented a 6% risk of death (1.060 95% CI: 1.034, 1.088). In 2009, there were 419 cases of SARI, 29 of which required endotracheal intubation and 14 of which died; 2 had laboratory results: 1 for pandemic influenza and 1 for influenza A. In 2018, there were 580 cases of SARI, of which 32 required endotracheal intubation, of which 17 died: 8 with influenza A(H1N1)pdm09 and 1 with influenza B Yamagata. All of these patients developed pneumonia. The risk increased when the patient was exposed to invasive mechanical intubation (2.379 95% CI 1.390, 4.072). The risk of death increases as the duration of the disease increases. Cough was the main indicator of influenza in both periods; in addition to these clinical data, fever and headache were also significant during the 2009 pandemic (Table 3). The protective effect of the influenza vaccine was highlighted in 2009.

When the multivariate model was constructed with the risk factors and symptoms, fever, cough and coryza remained statistically significant. An analysis of the deaths revealed that more than 50% of the deaths occurred in people over 45 years of age during 2018, whereas in 2009, more deaths occurred in people less than 24 years of age (Table 4).

The main risk factor for death from influenza was endotracheal intubation (RR = 3.87, 95% CI, 1.45 to 10.32) in 2018. Hospitalization had an OR = 1.16 (95% CI 1.10 to 1.23) in 2018 and an OR = 1.08 (95% CI 1.02 to 1.15) in 2009 (Table 4).

Three types of clinical severity data were associated with the risk of death (cyanosis, polypnea and dyspnea) in both periods (Table 5). In the multivariate analysis, both dyspnea and endotracheal intubation remained associated with death (Table 6).

### 3.5. Percentage of Positivity

Of the 1431 laboratory-positive cases reported in 2009, 1371 were cases of influenza A(H1N1)pdm09, 56 were cases of influenza A, and the rest of the cases were untyped influenza A. Among the 573 confirmed cases reported in 2018, the predominant cases were influenza A(H1N1)pdm09 (89.4%), influenza B Yamagata (5.6%), influenza A(H3) (3.5%), influenza B Victoria (1.0%) and influenza B (0.5%). In both states, the predominant circulating strain was influenza A(H1N1)pdm09, with cocirculation of influenza B. However, in Quintana Roo, influenza A(H3) was present in the middle of the epidemic wave. Most of the time, the percentage of positive samples in Yucatan was maintained above 40 percent (Figure 2).

### 3.6. Case Fatality Rate and Survivors

A total of 580 SARI hospitalizations and 49 inpatient SARI deaths (case fatality rate: 8.5% [95% CI: 6.449–10.99], *p* = <0.0000001) were reported to the IMSS system between weeks 21 and 37 of 2018. In Quintana Roo, there were 337 SARI hospitalizations and 25 inpatient SARI deaths (case fatality rate: 7.4% [95% CI: 5.075–10.72], *p* = <0.0000001). In Yucatan, there were 243 SARI hospitalizations and 24 inpatient SARI deaths (case fatality rate: 9.8% [95% CI: 6.727–14.27], *p* = <0.0000001). The case fatality rate among those confirmed cases was higher in 2018 (0.5 vs. 4.7 per 100 confirmed cases). Cases that were exposed to invasive mechanical ventilation had a lower probability of survival and a shorter time of survival compared to those who were not exposed (Figure 3).

## 4. Discussion

We characterized the epidemiology of a recurrent wave of influenza A(H1N1)pdm09 transmission on the Yucatan Peninsula spanning weeks 21 to 37 of 2018, on the basis of the ILI/SARI and laboratory-confirmed infections. We compared the impact, severity and age patterns. We used individual-level patient information collected through a prospective influenza surveillance system put into place especially for the 2009 pandemic by the largest Mexican Social Security medical system. We also documented a significant increase in the proportion of influenza A/H1N1 and ILI/SARI cases relative to the 2009 pandemic.

Previously, Australia experienced unusually high levels of influenza activity in 2017 because of several factors potentially contributing to the lower vaccine effectiveness, including the genetic diversity of the dominant influenza A(H3N2) strains currently circulating, the greater proportion of elderly affected individuals, who are known to have reduced responses to vaccines, and ongoing problems with identifying suitable influenza A(H3N2) vaccine candidates [8]. Intense interseasonal influenza outbreaks (in the Northern Hemisphere summer) have been documented in other countries and years [9,10,11] providing a clearer understanding of the epidemiology and burden of interseasonal influenza and trends over time. In this sense, Li et al. showed that the influenza virus had clear seasonal epidemics in winter months in most temperate sites, but the timing of epidemics was more variable and less seasonal with decreasing distance from the equator. The influenza virus had clear seasonal epidemics in both temperate and tropical regions but sometimes started in the late summer months in the tropics of each hemisphere; these patterns became less pronounced closer to the equator, with the emergence of summer epidemics in some sites [12].

Specifically, in this study, an increase was observed in ILI/SARI cases in the Yucatan Peninsula; it was due to influenza A(H1N1)pdm09 in both periods. Additionally, through sentinel epidemiological surveillance, the Ministry of Health of Mexico revealed an increase in cases of influenza in the Yucatan Peninsula, with a predominance of influenza A(H1N1)pdm09 circulation in 80% of the cases, where 59.9% of the cases were concentrated in Yucatan and Quintana Roo [2]. The Institute of Epidemiological Diagnosis and Reference (InDRE) has not identified mutations related to antigenic changes or changes in the virulence or pathogenicity of the influenza virus. No viral resistance to oseltamivir was identified in Mexico in this summer [2]. Similarly, in the IMSS, a substantial increase in the number of SARI hospitalizations during the period December 2011 to March 2012 and an older age distribution of laboratory-confirmed A(H1N1)pdm09 influenza hospitalizations and deaths relative to 2009 A(H1N1)pdm09 pandemic patterns were documented [13]. Additionally, the number of influenza A(H1N1)pdm09-related hospitalizations and deaths during the period of October 2013–January 2014 and the proportionate shift of severe disease to middle-aged adults relative to the preceding A(H1N1)pdm09 2011–2012 epidemic in the IMSS were reported [14]. Notably, in 2018, 66.5% of confirmed cases occurred in patients between 25 and 64 years of age, whereas in 2009, 61.8% were in patients between 5 and 24 years of age. At the end of 2010, 28.5 million doses were administered free of charge in Mexico [15]. After the 2009 influenza pandemic, programmatic groups (based on targets) were established in Mexico for influenza vaccination (so vaccination is not universal). Influenza vaccination in Mexico is based on the assumption that influenza peaks in winter. The target group included older adults and people from 19 to 59 years of age with risk factors; therefore, one hypothesis for the change in age observed in 2018 is that this target population received the vaccination. People who had received a seasonal influenza vaccine in the previous year had a reduced risk of influenza A(H1N1)pdm09 infection. Between October 2018 and March 2019, 10,902,605 doses of influenza vaccine were scheduled for the population insured by the IMSS; 160,090 doses were for Quintana Roo and 217,931 for Yucatan. The increase in cases during 2018 occurred prior to this vaccination event, so the hypotheses of this lag can be seen in the risk estimators for vaccination and the fatality rate in older adults. The discrepancies in the estimated risk of vaccination between the two periods are hypothesized to be that in 2009, vaccination was universal and in the intensive phase due to the pandemic year. In 2018, however, it was in target groups, and contradictorily, the point estimate was a modest risk; this can be explained by the vaccination history of the 2017–2018 season. Finally, in Mexico, the circulation of influenza A(H1N1)pdm09 coincided with a higher incidence of severe pneumonia, predominantly in young adults, in March and April 2009 [1]. The circulation of influenza viruses in Mexico after 2009 was biennial, in the winter [16]. What was observed in the interseasonal period of 2018 in the Yucatan Peninsula highlights that it was due to a sudden activity by influenza A(H1N1)pdm09, which has been characterized by abrupt increases each winter. Additionally, the interim influenza vaccine effectiveness (VE) estimates from Australia documented that the adjusted VE was low overall at 33% (95CI% 17 to 46) and 50% (95% CI: 8 to 74) for influenza A(H1N1)pdm09 [17]. In 2009, the protective factor of mass vaccination against influenza was observed throughout the year. However, this was not the case in 2018, when vaccination was targeted to certain programmatic groups.

Previously, it was documented that elderly participants also had a higher risk of hospital admission, and children aged less than 5 years were at a lower risk of death, probably because the children aged less than 5 years were at a higher risk of developing pneumonia [18].

In terms of clinical manifestations, dyspnea, tachypnea, cyanosis and being confined to bed were prognostic factors for hospital admission and death during influenza A(H1N1)pdm09 virus infection in Mexico in 2009 [19]; this was consistent with the findings of this study in 2009 and 2018 in a specific area of Mexico because the source of information is the same, although adjustments in the epidemiological surveillance system (diagnostic algorithms) have been made.

Asthma and obesity were identified as risk factors associated with the presentation of influenza in this study. Coleman BL et al. [20] also identified asthma as a risk factor for hospitalization, as well as obesity, especially in severe situations [21]. However, a meta-analysis of 59 studies revealed that obesity was a risk factor for death [18]. In the present study, obesity presented a minimal risk of death in both periods. It remains unclear whether obesity in itself is a risk factor or whether it reflects the presence of other comorbidities such as cardiovascular diseases and diabetes mellitus; in our risk model, obesity alone resulted in an HR of 1.55, with the limitation of analyzing more comorbidities simultaneously. This highlights the need to strengthen preventive measures in these groups, as well as adherence to asthma treatment. One risk worth highlighting was travel history in 2009, a year marked by the influenza pandemic, where travel history was given priority; this situation was not reflected in the estimated risks for 2018.

Although the phenomenon that occurred has been reported previously, this behavior is unique given that it is the first time that an abrupt increase of this magnitude has been observed in the months prior to the winter season since 2009 in Mexico. The influenza cases presented seasonality, with a single peak between the months of November and March of each year [3]. However, the Yucatan Peninsula (states of Campeche, Quintana Roo, and Yucatan) is one of the primary entry points for influenza variants in Mexico because of its large influx of tourists and commercial activities in the areas surrounding Merida and Cancun; population movement for recreational purposes in the summer or the behavior of the Southern Hemisphere influenza season could affect individuals traveling to the Peninsula. One hypothesis to consider is the climate in the region, with cold fronts generating a “typical temperate winter” [22], and the effects of the geographical location of the Yucatan Peninsula are determined mainly by the Caribbean Sea, resulting in a dry and warm winter and a summer with bimodal precipitation [23]. In this sense, although lower temperature was associated with higher activity of both the influenza virus and respiratory syncytial virus, higher relative humidity was associated with higher influenza virus activity when the temperature was above the annual average [12], and the geographical location of the Yucatan Peninsula is in the tropics, with high humidity; 85.5% of the state of Yucatan has a warm subhumid climate, while 99% of the state of Quintana Roo has a warm subhumid climate [24]. Another hypothesis to consider corresponds to previous studies on the historical behavior of the influenza virus in the southeastern region of the country, so modifying influenza vaccination schemes according to the different patterns observed in the southeastern region and the Yucatan Peninsula has been recommended [22], and one study reported that the seasonality of the virus is associated with the rainy season [25]. These findings are particularly important for the inhabitants of the Yucatan Peninsula, but they should also be considered for high-risk groups traveling to the region. Similarly, vaccination should be considered for use in confined populations, such as daycare and nursing homes and hospitalized persons who do not have a contraindication to the vaccine, where some susceptible groups will be found (for example, people with diabetes, cancer, immunosuppression and renal failure). In general, influenza dynamics in the tropical Yucatan Peninsula differs from the rest of the country [22]. Ayora has consistently insisted that assuming there is a single seasonal pattern for influenza in Mexico is a mistake, and therefore, determining whether there are different geographic patterns of virus and disease circulation is essential for developing and implementing control and prevention strategies [26].

The IMSS is a tripartite Mexican health system covering approximately 40% of the Mexican population and comprises workers in the private sector and their families, relying on a network of 1099 primary healthcare units and 259 hospitals nationwide [27].

In summary, our findings indicate a changing distribution of ILI/SARI and laboratory-confirmed A(H1N1)pdm09 influenza in the summer of 2018 relative to 2009 A/H1N1 pandemic patterns: the behavior of the epidemic curve, changes in age groups and the case fatality rate. This surprising behavior was documented in Australia during the summer of 2017. These authors reported that ILI activity was moderate during the interseasonal period from January to March of 2017; for example, on the Yucatan Peninsula, it presented a high percentage of positivity for several weeks.

The strengths of this study include the following: (1) it was a population-based study, (2) it had a regional focus, (3) it included confirmed cases, (4) it included a description of the percentage of positivity and (5) it included the time distribution of ILI/SARI cases during the different epidemic waves. Nevertheless, our findings need to be contextualized, considering several limitations. The analyzed data were derived from passive database systems, which collect information from patients who seek and receive medical care at IMSS healthcare facilities. Additionally, specific information about the administration of specific therapeutics before or during hospitalization was unavailable, and another limitation was not having the rates of vaccination by age group in 2017 and 2018 and comparing them with the information from the epidemiological surveillance system.

## 5. Conclusions

The dominant pattern of A(H1N1)pdm09 influenza activity on the Yucatan Peninsula in the summer of 2018 has not been observed since the influenza pandemic of 2009 in Mexico. This is the first time that such a drastic increase in influenza activity on the Yucatan Peninsula has been identified during the summer, which highlights the importance of maintaining epidemiological surveillance of influenza throughout the year.

## Figures and Tables

**Figure 1 microorganisms-13-01086-f001:**
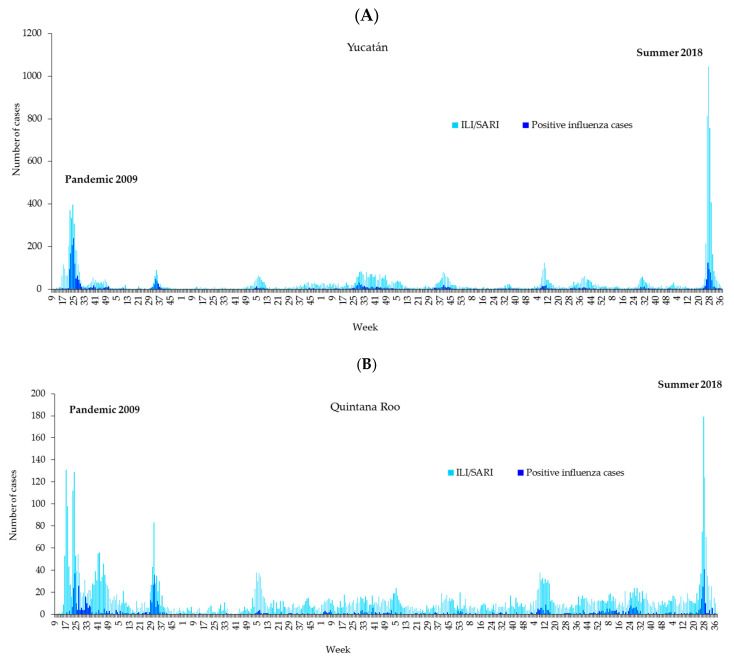
Epidemic curve of cases of ILI/SARI and positive results from the laboratory for influenza per week of onset of symptoms, 2009–2018. (**A**) Yucatan. (**B**) Quintana Roo.

**Figure 2 microorganisms-13-01086-f002:**
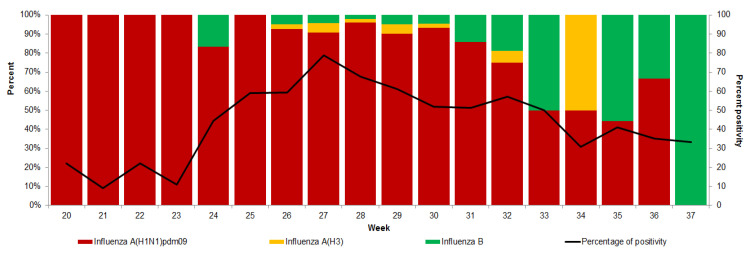
Percent positivity and results per week, Yucatan Peninsula, 20–37, 2018.

**Figure 3 microorganisms-13-01086-f003:**
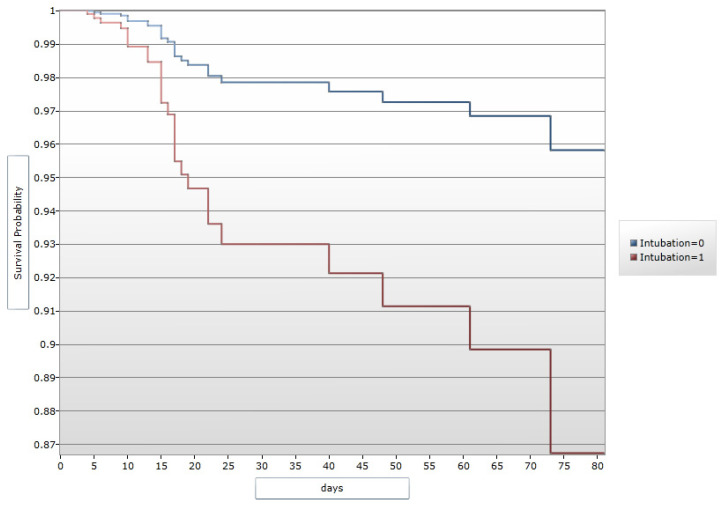
Kaplan–Meier Curve: survivors who were exposed to orotracheal intubation on the Yucatan Peninsula in 2018. 1: Yes. 0: No.

**Table 1 microorganisms-13-01086-t001:** Risk factors for contracting influenza, 2009 and 2018.

Variable	2009			2018		
	OR	95%CI		OR	95%CI	
Contact with animals	1.37	0.65	2.81	5.08	3.53	7.26
Vaccination	0.53	0.37	0.73	3.58	2.61	4.87
Immunosuppression	0.57	0.16	1.62	2.89	1.18	6.51
Diabetes mellitus	0.39	0.19	0.79	2.82	2.07	3.82
Obesity	0.63	0.27	1.34	2.55	1.88	3.41
Asthma	1.86	1.32	2.63	2.17	1.52	3.05
Smoking	0.53	0.22	1.17	1.92	1.17	3.08
HIV	0.76	0.17	2.69	1.82	0.52	5.22
COPD	0.13	0.01	0.69	1.59	0.64	3.51
Travel history	2.03	1.46	2.81	…	…	…

**Table 2 microorganisms-13-01086-t002:** Clinical manifestations associated with the presentation of influenza, 2009 and 2018.

Clinical Manifestations	2009			2018		
	OR	95%CI		OR	95%CI	
Cough	6.70	5.51	8.21	20.53	13.80	31.71
Fever	8.29	6.66	10.43	19.53	13.22	29.91
Coryza	0.93	0.60	1.41	6.32	4.57	8.73
Headache	4.98	4.24	5.88	6.23	4.93	7.94
Prostration	1.66	1.45	1.89	5.14	4.02	6.56
General discomfort	5.60	4.76	6.62	4.14	3.41	5.04
Sudden onset	2.78	2.43	3.18	3.84	3.17	4.66
Dyspnea	0.73	0.60	0.89	3.79	3.05	4.71
Arthralgias	2.34	2.06	2.65	3.36	2.79	4.05
Chest pain	1.36	1.14	1.62	3.19	2.55	3.98
Cyanosis	0.65	0.33	1.21	2.93	1.03	7.51
Polypnea	0.65	0.33	1.21	2.93	1.03	7.51
Rhinorrhea	2.55	2.23	2.91	2.85	2.37	3.42
Abdominal pain	1.15	0.96	1.39	2.58	1.99	3.32
Myalgias	2.52	2.22	2.86	2.52	2.22	2.86
Odynophagia	2.13	1.88	2.42	2.39	2.00	2.85
Diarrhea	0.97	0.78	1.22	2.38	1.81	3.11
Shivers	1.72	1.51	1.95	1.96	1.65	2.34
Conjunctivitis	1.72	1.49	1.99	1.59	1.20	2.10
Nasal congestion	2.11	1.85	2.40	…	…	…
Dysphonia	1.63	1.32	2.01	…	…	…
Low back pain	1.99	1.72	2.30	…	…	…

**Table 3 microorganisms-13-01086-t003:** Influenza risk model, 2009 and 2018.

Variable	2009			2018		
	OR	z	*p*	OR	z	*p*
Contact with animals	1.41	0.88	0.38	1.75	2.79	0.01
Vaccination	0.59	−2.90	<0.001	1.44	2.16	0.03
Immunosuppression	1.16	0.23	0.82	1.25	0.48	0.63
Diabetes mellitus	0.36	−2.69	0.01	1.50	2.45	0.01
Obesity	0.67	−0.95	0.34	1.12	0.71	0.48
Asthma	1.52	2.25	0.02	0.98	−0.09	0.93
Cough	2.93	9.34	<0.001	5.18	3.46	<0.001
Fever	3.59	10.15	<0.001	3.27	2.54	0.01
Coryza	0.63	−2.04	0.04	2.19	4.18	<0.001
Headache	2.09	7.74	<0.001	1.01	0.03	0.97
Prostration	1.00	−0.02	0.99	1.80	3.98	<0.001
Test	Statistic	D.F.	*p*-Value	Statistic	D.F.	*p*-Value
Score	651.50	11	<0.001	558.16	11	<0.001
Likelihood Ratio	792.76	11	<0.001	593.59	11	<0.001

**Table 4 microorganisms-13-01086-t004:** Risk factors for death from influenza, 2009 and 2018.

Variable	2009			2018		
	RR	95%CI		RR	95%CI	
Endotracheal intubation	1.39	0.87	2.23	3.87	1.45	10.32
Immunosuppression	1.33	0.75	2.34	1.27	0.85	1.90
Hospitalization	1.08	1.02	1.15	1.16	1.10	1.23
Obesity	1.12	0.87	1.48	1.14	1.03	1.26
Antibiotics	1.04	0.98	1.10	1.12	1.03	1.23
COPD	0.99	0.99	0.99	1.11	0.82	1.51
Diabetes mellitus	0.99	0.99	0.91	1.10	1.00	1.22
Chronic disease	1.02	0.99	1.06	1.08	1.01	1.14
Antiviral	0.99	0.99	1.00	1.07	1.04	1.09
Asthma	1.01	0.98	1.05	1.02	0.94	1.11
Contact with animals	1.09	0.919	1.29	0.95	0.93	0.97
Smoking	0.99	0.99	0.99	0.95	0.93	0.97
Vaccination	1.02	0.97	1.06	0.95	0.93	0.97
HIV	0.99	0.99	0.99	0.95	0.93	0.97
Pregnancy	0.99	0.98	0.99	0.95	0.93	0.97
Travel history	0.99	0.99	0.99	…	…	…

**Table 5 microorganisms-13-01086-t005:** Clinical manifestations associated with death from influenza, 2009 and 2018.

Clinical Manifestations	2009			2018		
	RR	95%CI		RR	95%CI	
Cyanosis	1.09	0.92	1.29	1.43	0.81	2.52
Polypnea	1.09	0.92	1.29	1.43	0.81	2.52
Dyspnea	1.03	1.00	1.07	1.20	1.12	1.30
Chest pain	1.01	0.99	1.02	1.11	1.04	1.18
Diarrhea	0.99	0.99	0.99	1.05	0.98	1.13
Prostration	0.99	0.99	1.00	1.04	0.98	1.10
Abdominal pain	0.99	0.99	0.99	1.04	0.97	1.10
General discomfort	1.00	1.00	1.01	1.04	1.01	1.07
Cough	1.00	1.00	1.01	1.01	0.92	1.09
Fever	0.97	0.93	1.01	1.01	0.93	1.09
Headache	0.99	0.98	1.01	1.00	0.95	1.05
Sudden onset	0.99	0.99	1.00	0.99	0.95	1.03
Odynophagia	0.99	0.99	1.00	0.99	0.96	1.03
Coryza	0.99	0.99	0.99	0.96	0.93	0.99
Arthralgias	0.99	0.98	0.99	0.96	0.93	0.99
Shivers	0.99	0.99	1.01	0.96	0.92	0.99
Rhinorrhea	0.99	0.99	1.00	0.95	0.91	0.99
Myalgia	0.99	0.99	1.00	0.93	0.88	0.98
Nasal congestion	0.99	0.99	1.00	…	…	…
Low back pain	0.99	0.99	0.99	…	…	…
Dysphonia	1.00	0.99	1.01	...	…	…

**Table 6 microorganisms-13-01086-t006:** Influenza death risk model, 2009 and 2018.

Variable	2009			2018		
	HR	Z	*p*	HR	Z	*p*
Endotracheal intubation	111.8	3.21	0.00	3.32	2.63	0.01
Obesity	0.01	−2.19	0.03	1.55	1.06	0.29
Dyspnea	1.56	0.30	0.76	40.16	3.57	0.00
Chest pain	1.03	0.04	0.97	0.83	−0.45	0.66
	Likelihood ratio test = 7.9746 on 4 df, *p* = 0.0925			Likelihood ratio test = 56.88 on 4 df, *p* < 0.001		
	n = 1431, number of events = 7			n = 573, number of events = 27		

## Data Availability

https://doi.org/10.6084/m9.figshare.28635506.v1 (accessed on 20 March 2025).

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
