# Peer review of "Characterization of a Novel 2018 Influenza Virus Outbreak on the Yucatan Peninsula, Mexico, in the Summer"

_microorganisms, 2025, doi:10.3390/microorganisms13051086_

Round 1
Reviewer 1 Report
Comments and Suggestions for Authors
The manuscript presents a robust retrospective epidemiological analysis of an unusual summer influenza A(H1N1)pdm09 outbreak on the Yucatan Peninsula, Mexico, in 2018. The authors effectively compare this event with the 2009 pandemic, demonstrating key differences in epidemiological patterns, age distributions, clinical severity, and case fatality rates. The study is significant, particularly for influenza surveillance strategy and public health preparedness in tropical/subtropical regions with unique seasonal behaviors.
Minor comments:
- Clarity and Language:
The manuscript would benefit from professional language editing to correct grammar, punctuation, and awkward phrasing.
There are instances of redundant expressions and inconsistent formatting (e.g., spacing, line breaks, use of symbols).
- Manuscript Structure:
The Results section is overly dense and contains detailed tables that may be more appropriate as supplementary material.
Several figures and tables are not clearly visualized(e.g., Figures 1–3, Tables 1–8).
- Interpretation of Findings:
The paper suggests hypotheses (e.g., effect of climate, vaccination timing) but does not test them directly or support them with supplementary data (e.g., vaccination coverage by age/year).
- Others minor issues:
Typo: “prostation” should be corrected to “prostration.”
Author Response
Minor comments:
Clarity and Language:
The manuscript would benefit from professional language editing to correct grammar, punctuation, and awkward phrasing.
There are instances of redundant expressions and inconsistent formatting (e.g., spacing, line breaks, use of symbols).
A = The change has been made to the manuscript.
Manuscript Structure:
The Results section is overly dense and contains detailed tables that may be more appropriate as supplementary material.
A = The tables were reordered. See pages 5, 6, 7, 8, 9, 15, 16 and 17.
Several figures and tables are not clearly visualized (e.g., Figures 1–3, Tables 1–8).
A = The figures were redacted and the tables reordered. Se pages 4 and 10.
Interpretation of Findings:
The paper suggests hypotheses (e.g., effect of climate, vaccination timing) but does not test them directly or support them with supplementary data (e.g., vaccination coverage by age/year).
A = More information on this subject was included in the discussion. See pages 11 and 12.
Others minor issues:
Typo: “prostation” should be corrected to “prostration.”
A = Thank you, the change has been made in the table.
Reviewer 2 Report
Comments and Suggestions for Authors
This manuscript presents a retrospective analysis of the 2018 influenza outbreak in the Yucatán Peninsula, Mexico, using data from four molecular diagnostic laboratories affiliated with the Mexican Social Security Institute. The study explores the outbreak from multiple perspectives, including patient demographics, geographic distribution, disease severity, influenza positivity, seasonality, and case fatality rates. Statistical tools such as odds ratios, Kaplan–Meier survival analysis, and Cox multivariate models were employed to assess disease risk and outcomes. The authors also compared the 2018 outbreak with the 2009 influenza pandemic and found that the 2018 epidemic was shorter but more expansive, with a higher number of cases and hospitalizations. They conclude that the summer 2018 influenza A(H1N1)pdm09 activity in the Yucatán Peninsula followed an unusual pattern not observed since the 2009 pandemic. However, there are several unclear statement and conclusion in this manuscript. Please find the below comments and concerns:
- The data and evidence presented in this manuscript are insufficient to support the conclusion that the 2018 influenza outbreak in the Yucatán Peninsula was "unique." As the analysis primarily focuses on comparing the 2018 outbreak with the 2009 influenza pandemic in Mexico, the results can only substantiate that the 2018 outbreak differed from the 2009 event within the same country. To justify the claim of uniqueness, broader data and comparisons are necessary—ideally including other influenza outbreaks both within Mexico and globally. Aside from the 2009 comparison, the only other reference made is to vaccine effectiveness data from Australia, which is not enough to establish uniqueness. Overall, limiting comparisons to a single prior outbreak within the same geographic context does not provide strong support for such a definitive conclusion.
- Several statements in Section 3.2 lack supporting data or figures. For instance, the authors claim that Quintana Roo data represents a "W-shaped" curve and that the most affected group in Yucatán was the infant population. However, no figures or data are provided to substantiate these claims. The only data presented in Section 3.2 is Table 1, which does not differentiate between Quintana Roo and Yucatán. Additionally, other statements—such as 17.6% of cases occurring in children under 9 years old in Quintana Roo and 13.6% in Yucatán, as well as 11.6% and 3.7% of cases in older adults, respectively—are made without citing any data source. Furthermore, the term “older adult” is not defined, which limits the clarity and interpretability of the findings.
- Section 3.3 lacks supporting data sources or figures, which makes it difficult to interpret and validate the results. Without visual representations or detailed tables, the narrative remains unsubstantiated. While the section summarizes differences between Yucatán and Quintana Roo, it does not clearly specify whether the data presented refer to the 2018 outbreak or the 2009 outbreak. Moreover, the analysis does not adequately explain how these regional differences support the conclusion that the 2018 outbreak was unique. To strengthen this conclusion, additional data, clearer context, and more comprehensive comparative analyses are necessary.
- In Section 3.4, the authors state that asthma was the main risk factor for influenza in 2009. However, according to Table 2, travel history shows the highest OR and CI 95% , suggesting it may have been a more significant risk factor. Additional clarification is needed to explain why asthma is highlighted as the primary risk factor over travel history. Furthermore, travel history data is not provided for the 2018 influenza outbreak, and the manuscript does not explain why this data is missing. Without comparable travel history data, the conclusion that animal contact was the main risk factor for the 2018 outbreak lacks sufficient support and is not fully convincing.
- Line 198-199: authors made the statement that the protective effect of the influenza vaccine was highlighted in 2009. However, there is not enough discussion and how data in Table 4 support this statement.
- Lines 220–221: The authors claim that Figure 2 demonstrates that both dyspnoea and endotracheal intubation remained associated with death. However, this conclusion is unclear, as Figure 2 only illustrates the positivity rates of different influenza virus types. There is no data or analysis in the figure that directly supports an association between these clinical features and mortality.
- Line 231 “56 were cases of influenza A”: please specify whether this refers to another subtype of influenza A or it is refers to influenza B.
- Line 237-238: Authors made the statement that Figure 3 shows the percentage of positive samples in Yucatan was remained above 40 percent. However, this information cannot be found in Figure 3. Figure 3 shows survival probability instead of positivity percentage.

Author Response
The data and evidence presented in this manuscript are insufficient to support the conclusion that the 2018 influenza outbreak in the Yucatán Peninsula was "unique." As the analysis primarily focuses on comparing the 2018 outbreak with the 2009 influenza pandemic in Mexico, the results can only substantiate that the 2018 outbreak differed from the 2009 event within the same country. To justify the claim of uniqueness, broader data and comparisons are necessary—ideally including other influenza outbreaks both within Mexico and globally. Aside from the 2009 comparison, the only other reference made is to vaccine effectiveness data from Australia, which is not enough to establish uniqueness. Overall, limiting comparisons to a single prior outbreak within the same geographic context does not provide strong support for such a definitive conclusion.
A = The order of the discussion paragraphs was changed, adding 10 references to the manuscript.
Several statements in Section 3.2 lack supporting data or figures. For instance, the authors claim that Quintana Roo data represents a "W-shaped" curve and that the most affected group in Yucatán was the infant population. However, no figures or data are provided to substantiate these claims. The only data presented in Section 3.2 is Table 1, which does not differentiate between Quintana Roo and Yucatán. Additionally, other statements—such as 17.6% of cases occurring in children under 9 years old in Quintana Roo and 13.6% in Yucatán, as well as 11.6% and 3.7% of cases in older adults, respectively—are made without citing any data source. Furthermore, the term “older adult” is not defined, which limits the clarity and interpretability of the findings.
A = The paragraph was reworded, and tables were added as supplementary material. See pages 4 and 5.
Section 3.3 lacks supporting data sources or figures, which makes it difficult to interpret and validate the results. Without visual representations or detailed tables, the narrative remains unsubstantiated. While the section summarizes differences between Yucatán and Quintana Roo, it does not clearly specify whether the data presented refer to the 2018 outbreak or the 2009 outbreak. Moreover, the analysis does not adequately explain how these regional differences support the conclusion that the 2018 outbreak was unique. To strengthen this conclusion, additional data, clearer context, and more comprehensive comparative analyses are necessary.
A = Modifications were made to the Results and Discussion sections, adding references. As with reviewer 1, a comment regarding the rearrangement of tables was addressed. See pages 4 and 5.
In Section 3.4, the authors state that asthma was the main risk factor for influenza in 2009. However, according to Table 2, travel history shows the highest OR and CI 95%, suggesting it may have been a more significant risk factor. Additional clarification is needed to explain why asthma is highlighted as the primary risk factor over travel history. Furthermore, travel history data is not provided for the 2018 influenza outbreak, and the manuscript does not explain why this data is missing. Without comparable travel history data, the conclusion that animal contact was the main risk factor for the 2018 outbreak lacks sufficient support and is not fully convincing.
A = A paragraph was included in the discussion explaining more about this finding. See page 13.
Line 198-199: authors made the statement that the protective effect of the influenza vaccine was highlighted in 2009. However, there is not enough discussion and how data in Table 4 support this statement.
A = A paragraph was included in the discussion explaining more about this finding. See page 12.
Lines 220–221: The authors claim that Figure 2 demonstrates that both dyspnoea and endotracheal intubation remained associated with death. However, this conclusion is unclear, as Figure 2 only illustrates the positivity rates of different influenza virus types. There is no data or analysis in the figure that directly supports an association between these clinical features and mortality.
A = The order of the tables was modified and the figure numbers were shortened.
Line 231 “56 were cases of influenza A”: please specify whether this refers to another subtype of influenza A or it is refers to influenza B.
A = The text was modified, detailing the results. See pages 9 and 10.
Line 237-238: Authors made the statement that Figure 3 shows the percentage of positive samples in Yucatan was remained above 40 percent. However, this information cannot be found in Figure 3. Figure 3 shows survival probability instead of positivity percentage.
A = Thanks for the comment, we changed. See pages 10 and 11.
Round 2
Reviewer 2 Report
Comments and Suggestions for Authors
With all the improvements, the reviewer does not have additional comments.